# Digital Technology in Clinical Trials for Multiple Sclerosis: Systematic Review

**DOI:** 10.3390/jcm10112328

**Published:** 2021-05-26

**Authors:** Marcello De Angelis, Luigi Lavorgna, Antonio Carotenuto, Martina Petruzzo, Roberta Lanzillo, Vincenzo Brescia Morra, Marcello Moccia

**Affiliations:** 1Multiple Sclerosis Clinical Care and Research Centre, Department of Neuroscience, Reproductive Science and Odontostomatology, Federico II University of Naples, 80138 Naples, Italy; marcello.deangelis91@gmail.com (M.D.A.); carotenuto.antonio87@gmail.com (A.C.); martinapetruzzo@gmail.com (M.P.); robertalanzillo@libero.it (R.L.); vincenzo.bresciamorra2@unina.it (V.B.M.); 2Department of Medical, Surgical, Neurological, Metabolic and Aging Sciences, University of Campania “Luigi Vanvitelli”, 81100 Naples, Italy; luigi.lavorgna@policliniconapoli.it

**Keywords:** multiple sclerosis, clinical trial, digital technology, outcome measures

## Abstract

Clinical trials in multiple sclerosis (MS) have been including digital technology tools to overcome limitations in treatment delivery and disease monitoring. In March 2020, we conducted a systematic search on pubmed.gov and clinicaltrials.gov databases (with no restrictions) to identify all relevant published and unpublished clinical trials, in English language, including MS patients, in which digital technology was applied. We used “multiple sclerosis” and “clinical trial” as the main search words, and “app”, “digital”, “electronic”, “internet” and “mobile” as additional search words, separately. Digital technology is part of clinical trial interventions to deliver psychotherapy and motor rehabilitation, with exergames, e-training, and robot-assisted exercises. Digital technology has been used to standardise previously existing outcome measures, with automatic acquisitions, reduced inconsistencies, and improved detection of symptoms (e.g., electronic recording of motor performance). Other clinical trials have been using digital technology for monitoring symptoms that would be otherwise difficult to detect (e.g., fatigue, balance), for measuring treatment adherence and side effects, and for self-assessment purposes. Collection of outcome measures is progressively shifting from paper-based on site, to internet-based on site, and, in the future, to internet-based at home, with the detection of clinical and treatment features that would have remained otherwise invisible. Similarly, remote interventions provide new possibilities of motor and cognitive rehabilitation.

## 1. Introduction

Multiple sclerosis (MS) is the most common cause of neurological disability in young adults [1]. MS symptoms encompass a wide range of neurological deficits including weakness, unbalance, visual disturbances, etc. [2]. A number of clinical features (e.g., fatigue, sensory symptoms, cognitive impairment, pain, and bladder dysfunction) are difficult to evaluate on clinical examination, and, thus, are frequently named invisible symptoms, but affect daily functioning, social life, close relationships, and leisure activities [3,4]. As such, MS patients require long-term, multidisciplinary care in both clinical and community settings [5], with the use of disease modifying treatments (DMTs) aiming at reducing the risk of relapses and, at least in part, halting disability progression, along with symptomatic treatments and rehabilitation [2].

Digital technology includes both the delivery of interventions and the collection of data (e.g., outcome measures), using a variety of digital tools, including smartphones, websites, apps, and wearable devices [6,7,8]. A number of health interventions have already leveraged the use of digital technology devices, methods and systems (e.g., mobile apps, digital health, e-health, internet-based interventions, etc.) [9,10,11,12], and have targeted patients, healthcare providers, health system and resource managers, and data services [9,13]. This would be particularly important for people with MS living in geographically isolated communities, who have difficulties in accessing rehabilitation services and have limited clinical trial opportunities [14,15,16,17,18]. As such, digital technology holds promise for remote interventions both for rehabilitation purposes (physical and cognitive) and to improve the overall clinical management, focusing on aspects that would be otherwise difficult to address in clinical practice [10].

Limitations of clinical outcome measures are widely known and could be possibly overcome by digital technology. The most common clinical outcome measures in MS clinical trials are the Annualised Relapse Rate (ARR) (i.e., relapse count over time), and the Expanded Disability Status Scale (EDSS) (a clinician-rated categorical scale ranging from 0 (normal neurological exam) to 10 (death resulting from MS)). However, clinical assessments in clinical trials and practice are usually run at specific intervals (e.g., 3 to 6 months), with the risk of missing mild relapses or subtle disease progression; indeed, more frequent consultations are frequently precluded by time, cost and geographical restraints [5,19]. Besides, traditional outcome measures do not reflect insidiously subtle progression of the disease and may fail to capture transient symptomatic and performance fluctuations affecting people with MS day-to-day [20]. For instance, the EDSS is clinically relevant and accessible to neurologists, but it is burdened by low intra- and inter-rater reliability at the lower end of the scale (e.g., early phases of MS) [21], and non-linear design at the upper levels, assigning a relatively large weight to walking abilities when compared with upper limb function [22]. While DMTs have proven anti-inflammatory activity (e.g., on ARR), the effect on EDSS is negligible, suggesting these measures lack of sufficient power to detect significant treatment effect [23,24,25]. Also, the EDSS does not capture invisible symptoms of MS [26]. More recently, to overcome these limitations, several composite outcome measures were developed, including both motor and cognitive tests, aiming to provide more complete and sensitive-to-change assessments [27]. For instance, the Multiple Sclerosis Functional Composite (MSFC) includes leg function by walking a short distance (“Timed 25-Foot Walk”, T25FT), arm function using peg-board test (“9-Hole Peg Test”, 9HPT), and attention/concentration test (“Paced Auditory Serial Addition test”, PASAT). Still, composite measures are time consuming, and do not account for some MS symptoms. Not least, in the past years, clinical trials have been using patient-reported outcome measures (PROMs) to estimate common and disabling symptoms, such as fatigue, depression and pain, which are difficult to detect on clinical consultation in the absence of standardized data gathering [27,28,29,30]. Thus, more precise clinical measures are urgently needed [31].

Considering limitations of conventional interventions and clinical measures, digital technology has become increasingly used to capture the complexity of MS in clinical practice, observational studies and clinical trials [32,33]. We specifically decided to focus on the use of digital technology in clinical trials, that are a turning point for outcome measures to move to clinical practice. For instance, the lesion count on brain MRI was developed in early 90s [34], and, then, widely applied in clinical trials, before becoming clinical practice [27]. Similarly, brain atrophy and neurofilament have been initially developed on clinical trial populations, and are now progressively becoming clinical practice [23,27,35,36]. Thus, studying current advancements in digital technology in clinical trials will help the understanding of possible future changes in MS clinical practice.

In the present review, we aim to—(1) systematically review concluded and ongoing MS clinical trials using digital technology (e.g., apps, mobile devices, electronic tools, internet, etc.) as either an intervention, or an outcome measure; and (2) narratively discuss how digital technology has been changing MS clinical trial design and outcome measures, along with limitations and future perspectives.

## 2. Materials and Methods

In March 2020, we conducted a systematic search on two different databases (pubmed.gov for published studies and clinicaltrials.gov for unpublished/ongoing studies); no search restrictions were applied (e.g., year range, country). We set “multiple sclerosis” and “clinical trial” as the main search words, and “app”, “digital”, “electronic”, “internet” and “mobile” as additional search words, separately. We identified all relevant published and unpublished/ongoing clinical trials and observational studies registered as clinical trials (e.g., NCT number). Following search results, we included only studies in the English language, and excluded reviews and other manuscripts. All search outputs (i.e., abstract, full paper, study design) were reviewed by two independent reviewers (MDA and MM) for consistency with study objectives and relevant documents were selected.

## 3. Results

We included thirty-five clinical trials—fifteen phase 2 clinical trials, one phase 3 clinical trial, and one phase 4 clinical trial. We also included sixteen observational studies which were registered on clinicaltrials.gov. Included studies are reported in Table 1. We excluded four studies that did not include any digital technology tool (Figure 1).

Looking at clinical trial inclusion criteria, studies included patients of any age (from 5 years, upwards). MS was the only disease in 31 out of 35 studies (88%), while the remaining 4 (12%) also included other neurological conditions, such as Parkinson’s disease, essential tremor, fibromyalgia, spinal cord injury, and stroke. Looking at clinical phenotypes of MS, six studies out of 35 (16%) only included relapsing-remitting MS, four studies (12%) included progressive MS, 22 studies (63%) included all phenotypes of MS, and three studies (9%) included clinically isolated syndrome (CIS) along with other MS subtypes.

From the preliminary review of selected studies, digital technology in MS clinical trials had three main purposes—testing digital technology as an intervention, standardising previously existing outcome measures, and detection of invisible clinical and treatment features (Figure 2).

### 3.1. Main Intervention

We found 10 ongoing studies out of 35 (29% of included studies) using digital technology as the main intervention. In particular, digital technology has been used as the main intervention for rehabilitation and psychotherapy. 

A number of studies are evaluating the possibility to improve adherence and effectiveness of rehabilitation using game-based virtual reality exercises on motor function [45], fatigue [46], and processing speed and attention in MS [41]. There are also ongoing studies evaluating the effects of internet e-training interventions on spasticity [40], and on general motor performances in MS patients [38]. A clinical trial is currently evaluating the role of social networks to improve adherence to exercise programs [42]. 

Two different studies are currently implementing tailored robot-based rehabilitation interventions. In particular, authors are testing the software for a weight support robot for gait training [39], and for a self-adapting robot for arm training in MS [37]. 

There are ongoing clinical trials studying feasibility and effects of web-based psychotherapy in MS. The POWER@MS1 study, a randomised controlled clinical trial, is evaluating the effects of a 2-year online behavioural intervention on DMT decision making, disease management and lifestyle, and on subsequent changes in inflammatory activity of MS [43]. The iSLEEPms study, a pilot randomised controlled trial, is evaluating the effects of a 4-week online cognitive-behavioural intervention on MS patients with sleep disorders [44].

### 3.2. Standardising Previous Outcome Measures

We found eight studies out of 35 (23% of included studies) aiming to standardise and improve reliability of outcome measures already used in clinical trials, among which two are already published. In particular, digital technology has been used to standardise and improve reliability of the EDSS, and, more in general, of composite neurological and cognitive examination in MS. 

Looking at the EDSS, one study is comparing conventional EDSS (paper version) with a new EDSS app [52]. In addition, in the EXPAND clinical trial, the EDSS, after being performed on site using Neurostatus E-Scoring App on an iPad (a companion app for this treatment development), was submitted online for remote revision. The Neurostatus E-Scoring App on iPad improved EDSS inter-rater reliability (with a reduction of inconsistencies from 32% to 1.5% during study conduction), and time spent in data collection (around 10 min per assessment) [71].

Looking at composite outcome measures, Rhodes and colleagues showed that the Multiple Sclerosis Performance Test (MSPT), a digital assessment tool built upon the previously-validated MSFC, allows the collection of standardised, quantitative, and clinically meaningful data in a clinical setting for individual assessments of patients with MS [49,72]. Similarly, in a sub-study of the phase 2 clinical trial of lisinopril in MS, conventional in-person MSFC has been compared with the mobile MSFC (a companion app for this treatment development), which is completed from the participants’ home using remote sensing technology and video conferencing [48].

Looking at upper and lower limb motor function, accelerometers on armbands have been compared with conventional measures of hand dexterity and ambulation in MS (e.g., 9HPT and T25WT, respectively) [50]. In another study, a newly developed App which performs hand dexterity tests is compared with the “Arm Function in Multiple Sclerosis Questionnaire”, a unidimensional 31-item questionnaire for measuring arm function in patients with MS, and with conventional hand exercises [51].

Finally, other studies have been evaluating EDSS-equivalent digital technology tools. In a proof-of-concept clinical trial, Midaglia and colleagues showed high adherence and satisfaction rates for the FLOODLIGHT test battery, including both active tests and passive monitoring (sensor-based gait and mobility recording), using smartphones and smartwatches [20]. Another study is assessing the efficacy of the BeCare app on smartphones to perform routine assessments of neurological functioning in subjects with MS; the BeCare app assigns EDSS scores obtained through multimodal information analysed with artificial intelligence techniques [54]. The same app is also been used to evaluate different neurological domains separately (cognition, afferent visual functioning, motor functioning, fine motor functioning, coordination, gait and endurance) [53].

### 3.3. Detection of Invisible Clinical and Treatment Features

We found 17 studies out of 35 (48% of included studies) aiming to improve detection of invisible clinical and treatment features based on digital technology, among which four are already published. In particular, digital technology has been used for daily monitoring of symptoms that would be otherwise difficult to detect, of treatment adherence and side effects, and for self-assessment purposes.

Looking at the invisible symptoms of MS, a number of ongoing studies are evaluating mobile Apps for daily monitoring of balance, falls [47,61], gait, fatigue and mood [57,58]. In a clinical trial designed to evaluate the effects of ginseng on MS-related fatigue, the authors developed the Real-Time Digital Fatigue Score (a companion app for this treatment development). This tool collects fatigue severity on a visual analogic scale (ranging from 0 to 10), through alarmed wristwatch device, worn by the patient, to measure and digitally record real-time experience of fatigue. Though the clinical trial failed to show significant treatment effect, the Real-Time Digital Fatigue Score has increased the statistical power for fatigue measurement [55]. 

Looking at treatment monitoring, in an already concluded clinical trial, the PROmyBETA App (a companion app for this treatment) was connected to electronic auto-injectors, and showed improved persistence, compliance, and adherence to interferon-beta1b over 6 months [59,65]. More recently, the PROmyBETA App has been integrated with a cognitive training tool designed with gaming elements, which could further contribute to improvements in interferon-beta1b usage [68]. In another ongoing trial, an electronic measure of needle disposal is used to measure adherence over two years for patients using injection therapy with glatiramer acetate [63]. Looking at oral medications (e.g., fingolimod, dimethyl fumarate, or teriflunomide), there is an ongoing trial evaluating an app connected to electronic pill bottles for monitoring medication usage and adherence over 90 days (time of bottle disposal) [70]. An electronic diary for MS patients, with medical data, patient reports, medication diaries and reminders, collected within a year of regular care, is under evaluation to measure and possibly improve adherence to different DMTs [60]. In line with this, Defer and colleagues showed that patients using the My eReport France^®^ App were more precise in reporting on adverse drug reactions, when compared with spontaneous patient reports [64]. Another study is evaluating the impact of nurse training on MS patients in improving reports through this app [69]. However, in another study including patients affected by different neurological diseases and different behavioural and lifestyle interventions, the authors found no significant difference over 6 weeks between a paper and electronic diary for monitoring adherence to behavioural and lifestyle suggestions given to the patient in a clinical setting [73].

Looking at self-assessment tools, one study is currently evaluating a mobile program called Digital Self-Assessment for Multiple Sclerosis, composed of walking, coordination, attention, and visual tests [66]. This app is currently tested also to evaluate the association with disability measured by neurologist on MSFC [62]. Similarly, in a large study including 633 progressive MS patients treated with Ocrelizumab, authors are using a digital tool (SymptoMScreen, (a companion app for this study)) to measure walking, spasticity, cognition, vision, fatigue, hand function, pain, sensory, bladder control, balance and depression/anxiety [67].

New outcome measures could come also from nanotechnologies, which could detect new respiratory biomarkers of inflammation and neurodegeneration of MS through an artificial olfactory system [56].

## 4. Discussion

In the present systematic review, we showed how clinical trials in MS have been leveraging digital technology to improve delivery, adherence and effectiveness of motor rehabilitation and psychotherapy, to standardise collection and interpretation of already existing outcome measures, and to detect MS clinical and treatment features that would otherwise remain invisible in clinical practice, also with the development of new outcome measures. With a number of digital technology tools being developed, the importance of people with MS has emerged, in order to build the technology around the patients, rather than having them fit into the technology.

Digital technology has been allowing remote interventions in MS, with specific regard to rehabilitation and psychotherapy. Rehabilitation has especially benefited from digital technology, with unprecedented possibilities of remote delivery and improved adherence, while maintaining the effectiveness of conventional in-person rehabilitation [74,75]. For instance, exergames in virtual reality settings combine physical exercise with leisure components of games [10]. Evidence for the use of exergames in MS will progressively emerge in the upcoming years, with a number of ongoing studies evaluating its effects on motor function [45], fatigue [46], cognition [41], and spasticity [40]. Looking at previous positive results in Parkinson’s disease and stroke [75,76,77], we expect that exergames in virtual reality settings will become standard tool for remote exercising in MS. Psychotherapy could also be delivered remotely and, for instance, there are ongoing clinical trials in MS evaluating the effectiveness of online cognitive-behavioural interventions on disease activity [44], and symptoms (e.g., sleep disturbances) [44].

The EDSS has been used in more than 120 phase 2 and 3 clinical trials in MS, but with very few medications showing treatment effect [27]. The possibility of acquiring the EDSS using digital tools is already well established [27,52]. Recently, in the EXPAND clinical trial, the EDSS was performed on site, using Neurostatus E-Scoring App on iPad, and, then, submitted online for centralized revision, with improved inter-rater reliability, possibly contributing to the positive results of clinical trial medication (siponimod) on EDSS progression and subsequent approval for secondary progressive MS [71]. In addition, digital technology could further refine disability outcome measures by combining the EDSS with more precise digital measures of upper and lower limb motor function (e.g., mobile Apps for hand dexterity tests, accelerometers, sensing technology, video conferencing, etc.) [48,50,51]. Thus, digital technology is ultimately enabling the development of combined outcome measures collecting standardised, quantitative, and clinically-meaningful data in a clinical setting for individual assessments of patients with MS, which could be easily moved from clinical trials to clinical practice [49,72]. Of note, in the case of the FLOODLIGHT test battery, which includes both active tests and passive monitoring, approximately 90% of patients wanted to see the results of their tests in real-time [20], suggesting patients were actually engaged.

Artificial intelligence and machine learning techniques will further contribute to improve the standard of care in MS. For instance, the BeCare app uses multimodal information analysed with artificial intelligence techniques to assign EDSS scores [54], and to separately evaluate different neurological domains (cognition, afferent visual functioning, motor functioning, fine motor functioning, coordination, gait and endurance), thus providing a detailed neurological examination and overcoming the weight lower limbs have on the EDSS [53]. Also, robot-based rehabilitation, which is already part of clinical practice [78], will further benefit from software improvements to tailor rehabilitation programs based on patient needs [37,39,78,79].

Patients’ empowerment has been a cornerstone in designing clinical trials with clinically meaningful outcome measures [27], and the possibility of real-time monitoring of MS patients has shed light on the most invisible aspects of the disease. There are a number of ongoing clinical trials evaluating mobile apps for daily monitoring of balance, falls [47,61], gait, fatigue and mood, ultimately improving the measurements and increasing the statistical power [55,57,58]. Multi-dimensional data collection of different MS disabilities through a mobile app could refine the quality and accuracy of knowledge on the disease progression, with improved standard-of-care of MS patients [66]. Though invisible symptoms are common to many neurological and non-neurological diseases (e.g., fatigue) [80], homogeneous populations and targeted interventions should be preferred. 

Non-adherence to DMTs is common, and is associated with higher risk of relapses and disability progression [81,82,83]. However, interventions to foster adherence in MS are costly, time-consuming, and difficult to evaluate in clinical practice [84,85]. Different studies have applied digital technology to auto-injectors [59,65], needle disposal systems [63], pill bottles [70], and diaries for monitoring DMT usage, adherence, and side effects [60,64].

Notwithstanding recent advances in digital technology in clinical trials (and daily life), there are limitations preventing the full applicability to clinical trials and clinical practice. First, digital technology access and use can be different between countries, due to cultural background, availability of high-speed connection, and trust in healthcare professionals [86]. Age is conventionally considered a limitation for digital technology, and, in line with this, most digital technologies in neurology have been applied in diseases affecting young adults, such as MS [87]. However, clinical trials we have hereby described included patients at any age, suggesting age-related limitations have been at least in part overcome [88]. Thus, a broader applicability would be very important for neurological diseases occurring in more advanced life stages (e.g., Parkinson’s disease, Alzheimer’s dementia), which could equally benefit from digital technology to detect their frequently subtle progression. Overall, clinical trial design should consider the actual possibility of using digital technology, along with possible bias coming from it (e.g., patients being aware of potential side effects of tested medications are more likely to present with placebo effect) [89]. The correct use of digital technology may be affected by the neurological condition itself (e.g., motor disability, visual impairment, psychiatric comorbidities, cognitive dysfunction) [88], thus limiting its applicability. The possibility of training a caregiver or a nurse to help with remote digital technology should be considered, though it has been tested only in one clinical trial [69]. There are financial limitations to the use of digital technology. Developing an app is relatively expensive, and requires a multi-step approach (i.e., device selection, app production, validation, data transfer) [90]. Most apps have been developed by private agencies that apply a fee, and, thus, at the moment, can only be used in selected population. Additional limitations are related to privacy and overall safety [89]. Finally, regulatory agencies do not frequently allow the use of outcome measures based on digital technology. The Food and Drug Administration has currently implemented a Digital Health Innovation Action Plan, which hopefully will enhance the potential of digital technology in clinical trials [91,92]. Similarly, the European Union has supported the Digital Transformation of Health and Care (Digicare), to develop a secure, flexible, and decentralised digital health infrastructure [93]. These initiatives will ultimately allow the standardisation of digital technology in clinical trials, observational studies, and clinical practice, enhancing its potentials while overcoming potential limitations [94].

Our systematic review has a number of limitations. Considering the relatively small number of included studies, we did not perform preliminary quality control, but only checked consistency with study objectives. Databases we have used (pubmed.gov, 26 May 2021, and clinicaltrials.gov, 26 May 2021) were not specifically organised for the topic of our review (e.g., absence of specific Medical Subject Headings), possibly limiting the number of included studies. Also, we need to acknowledge that a number of digital technology tools have been developed, but not included in clinical trials [93,95,96,97,98,99], and that digital health is a very dynamic and quickly evolving topic, with many studies being published after our research was carried out; thus, we have definitely missed some studies using digital technology in MS. However, we specifically focused on concluded and ongoing clinical trials (which are more likely to be translated to clinical practice) [27], while a full review of digital technology in MS should be evaluated in other studies.

## 5. Conclusions

With increasing numbers of MS patients using digital tools, especially following the COVID19 pandemic [100,101], the neurological and cognitive examination will be progressively shifting from paper-based on site, to internet-based on site, and, in the future, to internet-based at home. The latter could be particularly relevant since it allows continuous acquisition of real-world detailed outcome measures, moving beyond the limited snapshot of the neurological deficits assessed in clinical setting. As such, thanks to artificial intelligence algorithms, we could be able to detect relapses and disability automatically. Similarly, invisible symptoms of MS such as fatigue, cognitive impairment or anxiety/depression may be detected using digital technology, with the definition of totally new outcome measures, directly collected by patients in their real life, thus improving patients’ empowerment and participation into clinical trials.

MS is characterised by multiple symptoms (e.g., motor, sensory, cognitive, etc.) in common with other neurological disorders, and, thus, digital technology could be easily translated to other diseases. For instance, some clinical trials described above have already included patients with MS along with patients suffering from Parkinson’s disease, essential tremor, fibromyalgia, spinal cord injury, and stroke [37,39,73]. While there is no current digital technology standard in MS trials and research, over the next years, validation towards hard endpoints (i.e., EDSS 6.0 or EDSS 8.0) will possibly lead to a new universal language for monitoring disease outcomes and testing interventions.

In conclusion, digital technology is part of MS interventions and is widely used for cognitive and motor rehabilitation, with exergames, e-training, and robot-assisted exercises. Previously existing outcome measures are now collected using digital tools, in a reduced time frame, and with improved sensitivity. Specific platforms are available for the real-time detection of treatment adherence and side effects, and of symptoms which would have remained otherwise invisible, thanks to patients’ engagement and self-assessment. Of note, digital technology has been used in MS patients at any age, and with different clinical subtypes, suggesting it is able to tackle the spectrum of MS symptoms. Independently of its final goal, digital technology can develop a deep cultural change in clinical trial design and conduction, through tailoring studies and data collection on MS patients and positioning them at the very centre of the trial.

## Figures and Tables

**Figure 1 jcm-10-02328-f001:**
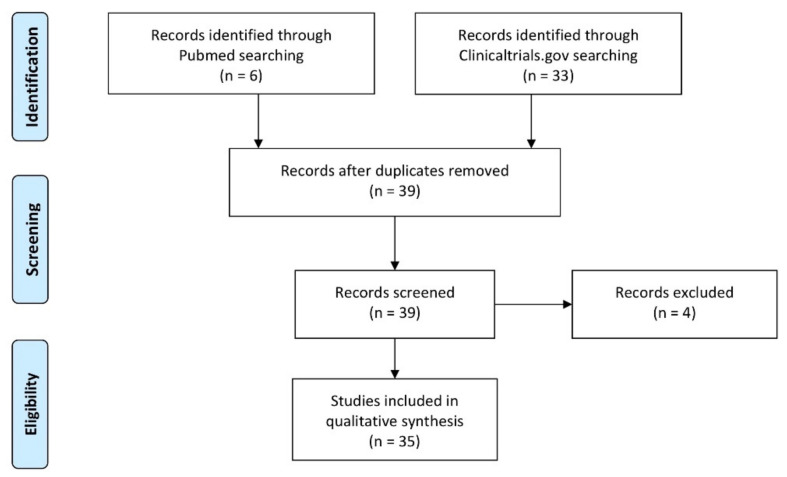
Study flow diagram.

**Figure 2 jcm-10-02328-f002:**
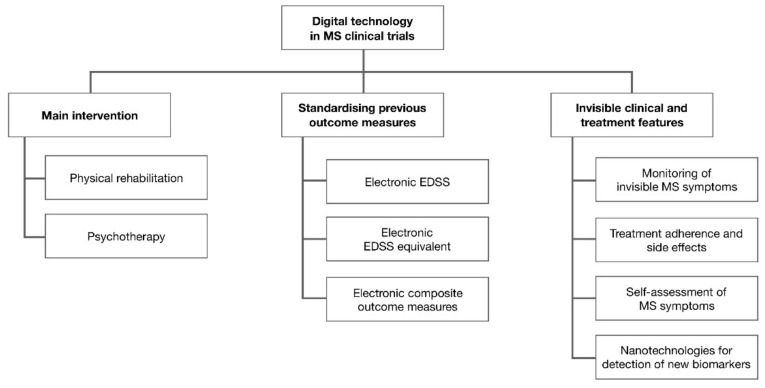
Digital technology in clinical trials for multiple sclerosis.

**Table 1 jcm-10-02328-t001:** Included studies.

Title	MS Phenotype	Other Conditions Included	Age	Study Design	Ref.
MAIN INTERVENTION					
Haptic feedback on robot (I-TRAVLE), arm strength and endurance training, 3 times a week for 8 weeks	All MS	Paralytic Stroke	>18	Interventional Open Label Single Group	[37]
Resistance-endurance training at home via internet (e-Training), 10–60 min	All MS		18–65	Interventional Randomized Parallel Assignment	[38]
Body-weight support robot (LEAP), usability study, therapist, observer and patient questionnaire	All MS	Spinal Cord Injuries Cerebral Palsy Parkinson Disease Stroke People with impaired lower extremity function	5–80	Interventional Open Label Single Group	[39]
Home-based movement training (MOTOmed), 12 weeks	All MS		>18	Interventional Randomized Parallel Assignment	[40]
Processing speed and attention treatment, digital Tablet-based game, 6 weeks	All MS + CIS	Cognitive Decline	18–71	Interventional Randomized Parallel Assignment	[41]
Internet-based intervention to increase physical activity (eFIT), 12 weeks	All MS		>18	Interventional Randomized Parallel Assignment	[42]
Web-based behavioral lifestyle intervention (POWER@MS1), 12 months	All MS		18–65	Interventional Randomized Parallel Assignment	[43]
Internet-delivered Cognitive Behavioral Intervention for Sleep Disturbance (iSLEEPms), 4 weeks	All MS		>18	Interventional Randomized Parallel Assignment	[44]
Game Based Virtual Reality Exercises (USE-IT) added to rehabilitation, 4 weeks	All MS		18–65	Interventional Randomized Parallel Assignment Open Label	[45]
(MORE STAMINA) Mobile App for Fatigue Management, adherence, 60 days	All MS	Fatigue Chronic Conditions	18–80	Observational Prospective	[46]
STANDARDASING PREVIOUS OUTCOME MEASURES					
Fall prevention exercise and education program with electronic diary, 8 weeks	All MS		18–89	Interventional Open Label Single Group	[47]
Comparison between Mobile Multiple Sclerosis Functional Composite and classic	Relapsing-remitting MS		18–65	Interventional Randomized Crossover Assignment Open Label	[48]
Usability of the Multiple Sclerosis Performance Test Device	All MS + CIS		>18	Observational Prospective	[49]
(Myo Armband) Wearable Biosensor to Track and Quantify Limb Dysfunction	All MS		18–65	Interventional Open Label Single Group	[50]
Change in the Arm Function in Multiple Sclerosis Questionnaire after 4 weeks of Home-Based Dexterity Training via Mobile App	All MS		18–75	Interventional Randomized Parallel Assignment	[51]
Automated EDSS Score Calculation Using a Smartphone Application vs paper version	Progressive MS		>18	Observational Prospective	[52]
Patient Centered Outcomes Analysis using BeCare Mobile App, 6 months	All MS		18–65	Observational Prospective	[53]
Validation of the BeCare Multiple Sclerosis Assessment App	All MS		18–75	Observational Prospective	[54]
DETECTION OF INVISIBLE CLINICAL AND TREATMENT FEATURES					
Realtime Digital Fatigue Score as outcome measure	All MS		18–70	Interventional Randomized Crossover Assignment	[55]
(NA-NOSE) Artificial olfactory system chemical sensor for the detection and identification of Multiple Sclerosis by Respiratory Samples	All MS		18–60	Observational Case-Only Cross-Sectional	[56]
Evaluation of adherence through BetaPlus Program elements (website)	Relapsing-remitting + Secondary Progressive MS		>18	Observational Prospective	[57]
Portuguese evaluation of adherence through BetaPlus Program elements (website)	Relapsing-remitting MS + Secondary Progressive MS		>18	Observational Prospective	[58]
Electronic measure of needle disposals (MEMS TrackCaps) as outcome measure	All MS + CIS		18–70	Interventional Randomized Parallel Assignment	[59]
Evaluation of adherence through the Use of an Electronic Diary, one year	All MS		18–70	Interventional Randomized Parallel Assignment Open Label	[60]
Fall Detection through Electronic Fall Detector, 6 months	All MS		>18	Interventional Randomized Parallel Assignment Open Label	[61]
Comparing a Smartphone Application with the Composite MSFC Score, 90 days	All MS		>18	Observational Prospective	[62]
Patient Reported Outcomes Measurement Information System via tablet, 6 weeks	All MS	Fibromyalgia Osteoarthritis Sjögren’s Syndrome Parkinson’s Disease	18–76	Interventional Randomized Parallel Assignment	[63]
Number of Active and Passive Tests Conducted by patients on smartwatch and smartphone	All MS		18–55	Interventional Non-Randomized Parallel Assignment Open Label	[20]
Change in number of reports of adverse drug reactions through app (My eReport France), 6 months	Relapsing-remitting MS		>18	Interventional Randomized Parallel Assignment Open Label	[64]
Betaferon adherence measured by mobile app (PROmyBETAapp), 6 months	Relapsing-remitting MS		>18	Observational Prospective	[65]
Digital Assessment Multiple Sclerosis 3 (DAMS-3) vs MSFC	All MS		18–60	Interventional Non-Randomized Sequential Assignment Open Label	[66]
Internet collection of Patient Reported Outcomes (SymptoMScreen) as outcome measure, 192 weeks	Progressive MS		18–65	Interventional Randomized Single-Group Assignment Open Label	[67]
Ascertaining Medication Usage & Patient Reported Outcomes Via the myBETAapp, 12 weeks	Relapsing-remitting MS		>18	Observational Prospective	[68]
Evaluation of the Impact of Patients’ Training by Nurse on Adverse Drug Reaction Reporting by Patient Via a Mobile Application (VIGIP-SEP2), 3 months	All MS		>18	Interventional Randomized Parallel Assignment Open Label	[69]
Promotion and evaluation of adherence via Electronic Pill Bottle, 90 days	Relapsing-remitting MS		>18	Interventional Randomized Parallel Assignement Open Label	[70]

## Data Availability

Not applicable.

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
