# Peer review of "Digital Technology in Clinical Trials for Multiple Sclerosis: Systematic Review"

_jcm, 2021, doi:10.3390/jcm10112328_

Round 1

Reviewer 1 Report

This article provides a broad systematic scoping review of the use of digital technologies in MS clinical trials. The topic is highly relevant to research and clinical care.  The authors outline the utility of digital technologies for capturing continuous and silent disease activity (both subjective and objective clinical data), for delivering interventions and for monitoring adherence. Thirty- five clinical trials met the inclusion criteria.  In this reviewers opinion this number is less than would be anticipated given systematic reviews for online exercise and/or rehabilitation management in MS are already published with at least this number of papers. Under the limitations sections - it is not a case of “may have missed” some papers – rather definitely missed published studies.  Authors should state so more clearly. Authors however importantly mention including trials in progress on clinicaltrials.gov to give readers a sense of future directions.  Digital technology has the potential to dramatically impact MS care.  This is especially true now with pandemic related policy further driving the virtual care agenda.  Digital technology examples mentioned include the Power@MS1 study, Floodlight and the BeCare MS app etc… There are many. 

The authors could consider a discussion of the limitations and benefits of many competing platforms – may the best ones rise to the top and be clinically implemented providing a new global universal language for monitoring disease outcomes and testing interventions?...  or will we be faced with no standard convention to compare across trials and time in MS research?  While not without limitations, hard end points with the EDSS has until now served this purpose (i.e. EDSS 6, 8)

The paper is balanced, pointing out the potential and the limitations of digital technology.  One reference to the later includes: “ different behavioural and life-style interventions, authors found no significant difference over 6 weeks between paper and electronic diary for monitoring adherence to behavioural and life-style suggestions given to patient in clinical setting [70].”  Thank you for the balanced viewpoints provided on pages 10-11.  I would recommend also adding to this section the importance of the technology being built around the patient, rather that than having patients “fit into the technology”. The last statement of the manuscript – placing the patient in the center – nicely ends with this priority. 

Thank you for the review.

Author Response

  • This article provides a broad systematic scoping review of the use of digital technologies in MS clinical trials. The topic is highly relevant to research and clinical care. The authors outline the utility of digital technologies for capturing continuous and silent disease activity (both subjective and objective clinical data), for delivering interventions and for monitoring adherence. Thirty- five clinical trials met the inclusion criteria. In this reviewers opinion this number is less than would be anticipated given systematic reviews for online exercise and/or rehabilitation management in MS are already published with at least this number of papers. Under the limitations sections - it is not a case of “may have missed” some papers – rather definitely missed published studies. Authors should state so more clearly. Authors however importantly mention including trials in progress on clinicaltrials.gov to give readers a sense of future directions. Digital technology has the potential to dramatically impact MS care. This is especially true now with pandemic related policy further driving the virtual care agenda. Digital technology examples mentioned include the Power@MS1 study, Floodlight and the BeCare MS app etc… There are many.

We thank the Reviewer for his/her suggestions. We have now clearly mentioned in the limitations’ section of the Discussion that: “we have definitely missed some studies using digital technology in MS”.

  • The authors could consider a discussion of the limitations and benefits of many competing platforms – may the best ones rise to the top and be clinically implemented providing a new global universal language for monitoring disease outcomes and testing interventions?...or will we be faced with no standard convention to compare across trials and time in MS research? While not without limitations, hard end points with the EDSS has until now served this purpose (i.e. EDSS 6, 8)

We agree this issue should have been better pointed out, and have now added the following sentences to the Conclusions: “While there is no current digital technology standard in MS trials and research, over the next years, validation towards hard endpoints (i.e., EDSS 6.0 or EDSS 8.0) will possibly lead to a new universal language for monitoring disease outcomes and testing interventions”.

  • The paper is balanced, pointing out the potential and the limitations of digital technology. One reference to the later includes: “ different behavioural and life-style interventions, authors found no significant difference over 6 weeks between paper and electronic diary for monitoring adherence to behavioural and life-style suggestions given to patient in clinical setting [70].” Thank you for the balanced viewpoints provided on pages 10-11. I would recommend also adding to this section the importance of the technology being built around the patient, rather that than having patients “fit into the technology”. The last statement of the manuscript – placing the patient in the center – nicely ends with this priority.

As suggested, we have now added the following sentence to the Discussion: “With a number of digital technology tools being developed, the importance of people with MS has emerged, in order to built the technology around the patients, rather that than having them fit into the technology”.

Reviewer 2 Report

This paper from De Angelis et al. aims at reviewing the literature about the use of digital technology in MS.

The use of digital tools in medicine is a very wide field and a topic of interest, especially for the management of chronic diseases like multiple sclerosis. Therefore, it can be relevant to propose a review article about this topic.

The article is globally well written and I have 2 minor comments

1) The systematic search in both databases have been conducted more than one year ago now and consequently some information may be outdated at the time of publication of the article as digital health is a. very dynamic and quickly evolving topic.

2) The authors aim at making a review about "Digital Technology in Clinical Trials" for MS, as written in the title. Digital Technology is a very wild term but the review mainly focuses on web based tools and mobile apps if we look at the keywords that were used for the systematic review. Additional keywords such as "wearable", "virtual reality", "serious games", or even "artificial intelligence", "machine learning", ... may have allowed to find a higher number of relevant studies. Consequently I think that this point should be clarified in either way (either being more precise in the manuscript title and objective regarding the field that was concerned by the study, or enlarging the review to be a little bit more exhaustive).

As a conclusion, i confirm that a review article regarding the currently published / ongoing studies employing digital tools in MS would be a good added value and so this article should be published after corrections.

Author Response

  • This paper from De Angelis et al. aims at reviewing the literature about the use of digital technology in MS. The use of digital tools in medicine is a very wide field and a topic of interest, especially for the management of chronic diseases like multiple sclerosis. Therefore, it can be relevant to propose a review article about this topic. The article is globally well written and I have 2 minor comments

We thank the Reviewer for his/her suggestions.

  • The systematic search in both databases have been conducted more than one year ago now and consequently some information may be outdated at the time of publication of the article as digital health is a very dynamic and quickly evolving topic.

As suggested by the Reviewer, we have now added the following sentences to the limitations’ section of the Discussion: “Digital health is a very dynamic and quickly evolving topic, with many studies being published after our research was carried out; thus, we have definitely missed some studies using digital technology in MS”.

  • The authors aim at making a review about "Digital Technology in Clinical Trials" for MS, as written in the title. Digital Technology is a very wild term but the review mainly focuses on web based tools and mobile apps if we look at the keywords that were used for the systematic review. Additional keywords such as "wearable", "virtual reality", "serious games", or even "artificial intelligence", "machine learning", ... may have allowed to find a higher number of relevant studies. Consequently I think that this point should be clarified in either way (either being more precise in the manuscript title and objective regarding the field that was concerned by the study, or enlarging the review to be a little bit more exhaustive).

We have been thinking a lot on the title and we feel that any other option would be limiting and less informative. As mentioned and references in the Introduction: “Digital technology includes both the delivery of interventions and the collection of data (e.g., outcome measures), using a variety of digital tools, including smartphones, websites, apps, and wearable devices”. Indeed, we have actually included virtual reality (ref. 38-42), wearable device (ref. 53), and artificial intelligence studies (ref. 55). However, we have now mentioned in the Discussion that: “We need to acknowledge that a number of digital technology tools have been developed, but not included in clinical trials…; thus, we have definitely missed some studies using digital technology in MS”.

  • As a conclusion, i confirm that a review article regarding the currently published / ongoing studies employing digital tools in MS would be a good added value and so this article should be published after corrections.

Thanks again for your supportive comments.